# Analysis of the p53 pathway in peripheral blood of retinoblastoma patients; potential biomarkers

**Mayra Martínez-Sánchez**[1], **Mariana Moctezuma-Dávila**[1], **Jesús Hernandez-Monge**[2],
**Martha Rangel-Charqueño**[3], **Vanesa Olivares-Illana**[1]*

1 Laboratorio de Interacciones Biomoleculares y Cáncer, Instituto de Física, Universidad Autónoma de San Luis Potosí, San Luis Potosí, SLP, México, 2 Catedra CONACyT, Laboratorio de Interacciones Biomoleculares y Cáncer, Instituto de Física, Universidad Autónoma de San Luis Potosí, San Luis Potosí, SLP, México, 3 División de Cirugía, Departamento de Oftalmología, Hospital Central "Ignacio Morones Prieto", Universidad Autónoma de San Luis Potosí, San Luis Potosí, SLP, México

* vanesa@ifisica.uaslp.mx

## Abstract

Loss of retinoblastoma (RB) function in the cone cells during retina development is necessary but not sufficient for retinoblastoma development. It has been reported that in the absence of RB activity, a retinoma is generated, and the onset of retina cancer occurs until the p53 pathway is altered. Unlike other types of cancer, in retinoblastoma the p53 tumour suppressor is mostly wild type, although its two primary regulators, MDMX and MDM2, are commonly dysregulated. A mutated RB form is inherited in around 35% of the cases, but normally two, somatic mutations are needed to alter the RB function. Here we investigated the mRNA levels *of RB*, *p53*, *MDMX* and *MDM2* in peripheral blood samples of retinoblastoma patients to monitor the pathway status of p53 in somatic cells. We sought to investigate the involvement of these genes in the development of retina cancer, with the aim of identifying biomarkers for early diagnosis of this disease.

## Introduction

Retinoblastoma is a common neoplasia in the pediatric population all around the world, being the second most common cancer in children under four years of age. The incidence of the disease varies depending on the country or the region. In developed countries, the rates are between 2.2 to 6.2 per million habitants, while in non-developed countries the rates can increase up to 24.5 per million habitants [1, 2]. Nevertheless, in Latin American countries, the incidence rate of the disease is underestimated due to the lack of a national retinoblastoma registry and systematic statistics. Currently, the diagnosis of retinoblastoma is mainly clinical. In non-developed countries, where incidence is higher, diagnosis is performed during the advanced stages of the disease, compromising the integrity of the eye and even the life of the children where the mortality rate is as high as 70% [1, 3–5]. The most commonly used therapeutic modalities include chemotherapy, radiation therapy, cryotherapy, thermotherapy,

**Data Availability Statement:** All relevant data are within the paper and its Supporting Information files.

**Funding:** This work was supported by Conacyt CB-256637. This work, including the efforts of Vanesa Olivares-Illana, was funded by L´oreal-UNESCO-AMC. JH-M Cátedras CONACyT program. The funders had no role in study design, data collection and analysis, decision to publish, or preparation of the manuscript.

**Competing interests:** The authors have declared that no competing interests exist.

enucleation (removal of the eyeball and optic nerve) and even orbital exenteration. The side effects of these treatments are important. Enucleation and exenteration, for example, have significant visual, cosmetic and psychological consequences due to the mutilation of the child.

Without treatment, mortality from this tumour is 99%. Gaining information about genetic background is required in people affected by the disease in order to improve the diagnosis and treatment. Due to the localisation and nature of this tumour, a biopsy is generally not a preferable option for the diagnosis of disease. With this in mind, we observe an urgent necessity to use non- or less-invasive methods for diagnosis. Blood samples represent a good option for the development of molecular diagnostic techniques; it has been used to search for early biomarkers of some diseases including hypothyroidism [6], non-small cell lung cancer [7] and Micro-RNAs in retinoblastoma [8].

Retinoblastoma results from cells lacking the *RB* gene, but in contrast with the majority of the cancers, p53 gene is not altered [9]. However, the p53 pathway has been implicated in the development of retinoblastoma in different ways. It was reported that in the 75% of cases the p53 pathway is altered, MDMX is upregulated in the 65% and MDM2 in the 10% of cases, even those demonstrating a wildtype p53 [3, 10]. Recently, it has also been shown that MDM2 but not MDMX promotes retina cancer in a p53-independent manner by regulating the translation of MYCN [11]. Some polymorphisms in the p53 pathway members have also been associated with the retinoblastoma development [12]. The importance of the p53 pathway in retinoblastoma was also shown in the mice with retina lacking RB and p107 and p53. Those mice developed bilateral retinoblastoma with 100% penetrance [13, 14]. The tumour arises in RB-depleted cone precursors, and the retinoblastoma proliferation depends on the cone precursors features such as high expression of oncoproteins like MDM2 [15]. The proto-oncoproteins MDM2 and MDMX are the primary p53 regulators. MDM2 is a ubiquitin ligase that, under normal cellular conditions, can ubiquitinate and degrade p53 via the proteasome 26S [16]. The MDM2-homologous protein MDMX shares high identity with MDM2, and despite this protein lacking ligase activity it binds p53 and MDM2 to aid in the p53 polyubiquitination process [17]. Retina precursor cells deficient in the *RB* gene suffers p53-dependent apoptosis. The overexpression of MDMX [18] or MDM2 genes promotes p53 inactivation and undergoes tumour progression and development [3, 10].

We hypothesise that these observations can also be detected in blood samples of the patients with retina cancer, and could be the results of hereditary causes. In the present work, we analysed *p53*, *MDMX*, *MDM2*, and *RB* mRNA expression levels in blood samples from retinoblastoma patients and family members, and compared them with healthy controls. The results presented here facilitates the utilisation of blood samples in retina cancer patients for Real-Time Quantitative Polymerase Chain reaction (RT-qPCR) experiments as a potential diagnostic tool.

## Materials and methods

### Ethics statement

The study was developed within the framework of the approved protocol number 12–17 by the committee of research and ethics from the Central Hospital "Ignacio Morones Prieto" San Luis Potosí, SLP, México. The ethics committee specifically approved this study. Diagnostic and therapeutic manoeuvres were carried out according to the Official Mexican Standard NOM-012-SSA3-2012, 2012 as well as with the current international codes for good practices in clinical research. The principles of the Helsinki Act of 1964 and its last revision in October 2013 were not transgressed. Written/signed informed consent was obtained from all the subjects participant in this study and the guardians on behalf of the minors involved in the study before

samples were collected and clinical data were obtained from all subjects during the interview and clinical examination.

## Sample collection

The sample of the study consisted of individuals recruited in ophthalmology clinic and healthy volunteers from the Central Hospital "Ignacio Morones Prieto" San Luis Potosí between August 2015 and November 2019. Peripheral venous blood samples were obtained from seventeen retinoblastoma patients, twenty healthy family member samples and twenty-seven healthy controls. Inclusion criteria: Children with diagnosis of retinoblastoma, without a known diagnosis of other cancer. Exclusion criteria: Patients with diagnosis of other type of cancer. Controls: Healthy children at pediatric age and healthy adults in the same range of age of the family member patients. Samples of 3 to 5 ml of whole blood were collected and processed by HISTOPAQUE-1077 (SIGMA. Cat. 10771) density gradient to isolate peripheral blood mononuclear cells (PBMC), 0.25mL of PBMC ($5x10^6$ cells) were lysate with 0.75mL of TRIzol reagent (Invitrogen, Thermo fisher. Cat 15596018) and stored at minus 80˚C until processed for RNA extraction. None of the patients included in the study had treatment at the sample collection time. Healthy controls were considered as individuals with no relevant medical history.

## RNA extraction

250 µl of mononuclear cells mixed with TRIzol were used to extract total RNA according to the manufacturer's protocol instruction. The precipitated RNA was resuspended in 50 µl of nuclease free water and treated with DNase I (RNase free, NEB-M0303S). The concentration and A260/280 ratio of purified RNA were measured between 1.7 and 2.0 using a Nanodrop 2000/2000c Spectrophotometer. RNA was stored at -80˚C until use.

## cDNA synthesis

The purified total RNA was retrotranscribed (1µg in each case) into cDNA in a reaction volume of 20 µl. The tubes were placed in the thermal cycler OptiMax Multigne TC9610 at 65˚C for 5 minutes and then put on ice. Next, 2 µl of 10-x M-MuLV Reverse Transcriptase buffer (NEB-B0253S), 1 µl RNAseOUT (Invitrogen-100000840); 1 µl M-MuLV Reverse Transcriptase enzyme- (New England Biolabs-M0253S) and 2 µl dNTP´s [10mM] (Invitrogen-10297-018) were added. Tubes were placed in the thermal cycler at 4˚C for 5 minutes, 60 minutes at 42˚C, 72˚C for 5 minutes and 4˚C for 15 minutes. Three repetitions were performed.

## Electrophoresis analysis of the amplicons

The six housekeeping genes (18S, TBP, B2M, HPRT, RPL13a, GAPDH) were PCR amplified under the following conditions: denaturing temperature 95˚C, melting temperature 60˚C, extension temperature 68˚C for 35 cycles, in a final volume of 25 µl. The reactions were analysed using 1.5% ultrapure agarose (Invitrogen-16500-100) gel in Tris-EDTA buffer pH = 7.4 (SIGMA-93302) at 80 V. The gel was stained with ethidium bromide (SIGMA-E7637) and photodocumented.

## Quantitative real-time PCR

Real-time quantitative PCR reaction was performed in 96-well microtiter plates using 7500 fast Real-Time PCR instrument (Applied Biosystems). The amplification mixture consisted of 1 µl of each primer [10 µM], 20 ng of the cDNA template and SYBR Green master mix

(Thermofisher-K0221) in a final volume of 12.5 μl. The PCR cycle conditions were set as follows: an initial denaturation step at 95˚C for 10 minutes followed by 40 cycles of 95˚C for 15 seconds, 60˚C for 30 seconds, and 72˚C for 30 seconds. Three replications were performed for each sample and each assay included a blank.

### Data analysis

The statistical analysis was performed using Graphpad Prism (version 6.0d GraphPad Software, CA). The expression stability of the potential housekeeping genes was evaluated by different methods. The coefficient of variation (CV) was used as a measure of expression stability. NormFinder is a method used to identify stably expressed genes among a set of housekeeping genes [19]. RefFinder is a user-friendly web-based comprehensive tool that calculates a geometric mean to assign a final ranking [20,21]. To examine the association between gene expression and various patient characteristics, analysis of variance (ANOVA). According to Shapiro-wilk test, the data showed a non-parametric distribution, then a Mann-Whitney test was used. The association of the expression between the different genes was assessed by Spearman´s correlation. P values < 0.05 regarded statistically significant.

## Results

### Study subjects

A total of 64 blood samples of retinoblastoma patients, healthy family members, and healthy controls were collected. Seventeen retinoblastoma patients participated in this study with the following characteristics: 59% male and 41% female, with a range between 0–6 years old (0–1, 41%; 1–6, 59%), 29% of patients present bilateral retinoblastoma, whereas 71% presented with unilateral retinoblastoma, the mean age of diagnosis was 4.5 ± 3.0 and 24.3 ± 21.86 months respectively, the characteristics of the patients are summarised in Table 1. No patients were under treatment at the time of the study. Twenty healthy family members of the patients and twenty-seven controls in the same range of ages were recruited.

### Choosing the housekeeping gene

We analysed six of the most commonly used housekeeping genes: β-2-microglobulin (B2M), Ribosomal protein L13a (RPL13A), Glyceraldehyde-3- phosphate dehydrogenase (GAPDH), Hypoxanthine phosphoribosyltransferase 1 (HPRT1), 18S Ribosomal RNA (18S), and TATA-binding protein (TBP) (S1 Table). The primers used are listed in Table 2, and amplicon sizes were checked on agarose gel electrophoresis (S1A Fig and Table 2). The analysis of the expression levels showed that the amplitude in the expression levels is higher in patient samples than in the healthy controls (S1B Fig), and also demonstrated that there were significant differences in B2M, RPL13a and GAPDH expression between the two groups. The stability of the potential housekeeping genes was examined by calculated the coefficient of variation (CV). The results showed that HPRT, TBP, and 18S are the optimal choices for an internal control gene (S2 Table and S1C Fig). Next, we tested 18S, TBP, and HPRT using the algorithm RefFinder [20]. According to RefFinder, HPRT, had a geometric mean of 1.32 and was ranked as the best housekeeping gene, as the lowest geometric mean value represents the most stable and better-ranked gene. HPRT was followed by TBP (geometric mean of 1.68), and finally 18S (geometric mean of 2.28) (S3 Table). Using the NormFinder program, a mathematical model based on the variance estimation that inversely correlates to the expression stability; the lowest stability value indicates the more stably expressed candidate genes [19]. NormFinder revealed that HPRT (0.016) and TBP (0.024), were the more stable housekeeping genes. These results were

**Table 1. Patient's characteristics.**

| Patient ID | Age at diagnosis (months) | Sex | Laterality | Mestastasis |
|---|---|---|---|---|
| 1 | 6 | M | B | No |
| 2 | 12 | M | U | No |
| 3 | 9 | F | U | No |
| 4 | 7 | M | U | No |
| 5 | 22 days | M | B | No |
| 6 | 19 | F | U | No |
| 7 | 6 | F | B | No |
| 8 | 2 | M | B | No |
| 9 | 18 | M | U | No |
| 10 | 8 | M | B | No |
| 11 | 60 | F | U | No |
| 12 | 20 | M | U | No |
| 13 | 23 | M | U | No |
| 14 | 6 | M | U | No |
| 15 | 8 | F | U | No |
| 16 | 36 | F | U | No |
| 17 | 74 | F | U | No |

corroborated using the geNorm software [21]; we calculated the M value, which is a reflection of the relative stability between the analysed reference genes [21]. The better-ranked genes using this approach were HPRT and TBP (M = 1.45), whereas 18S presented an M value much higher than the cut-off for geNorm (M = 4.13) (S3 Table). The following results were then analysed using the geometrical mean of HPRT and TBP.

**Table 2. Primers sequence used for the real time PCR assay.**

| Abbreviation | Primer sequence (5´to 3´) | Amplicon size (bp) | References |
|---|---|---|---|
| p53 | F: CGTCCCAAGCAATGGATGAT | 95 | This work |
| | R: TGGCATTCTGGGAGCTTCAT | | |
| MDM2 | F: AGATTGCAACAGATGTTGGGC | 119 | This work |
| | R: AGCCCTCTTCAGCTTGTGTT | | |
| MDMX | F: TCTGAGAGTGCTTGCAGGAT | 104 | This work |
| | R: AACATTTGACCTTGCGCACC | | |
| RB | F: GGGCGGAAGTGACGTTTTC | 95 | This work |
| | R: TCCCCTGAGAAAAACCGGAC | | |
| GAPDH | F: TCCAAAATCAAGTGGGGCGA | 115 | This work |
| | R: TGATGACCCTTTTGGCTCCC | | |
| HPRT1 | F: TGACACTGGCAAAACAATGCA | 94 | 13 |
| | R: GGTCCTTTTCACCAGCAAGCT | | |
| B2M | F: TGCTGTCTCCATGTTTGATGTATCT | 86 | 13 |
| | R: TCTCTGCTCCCCACCTCTAAGT | | |
| TBP | F: GAGCTGTGATGTGAAGTTTCC | 117 | 19 |
| | R: TCTGGGTTTGATCATTCTGTAG | | |
| RPL13a | F: CATAGGAAGCTGGGAGCAAG | 157 | 16 |
| | R: GCCCTCCAATCAGTCTTCTG | | |
| 18S | F: GGAGTATGGTTGCAAAGCTGA | 129 | 19 |
| | R: ATCTGTCAATCCTGTCCGTGT | | |

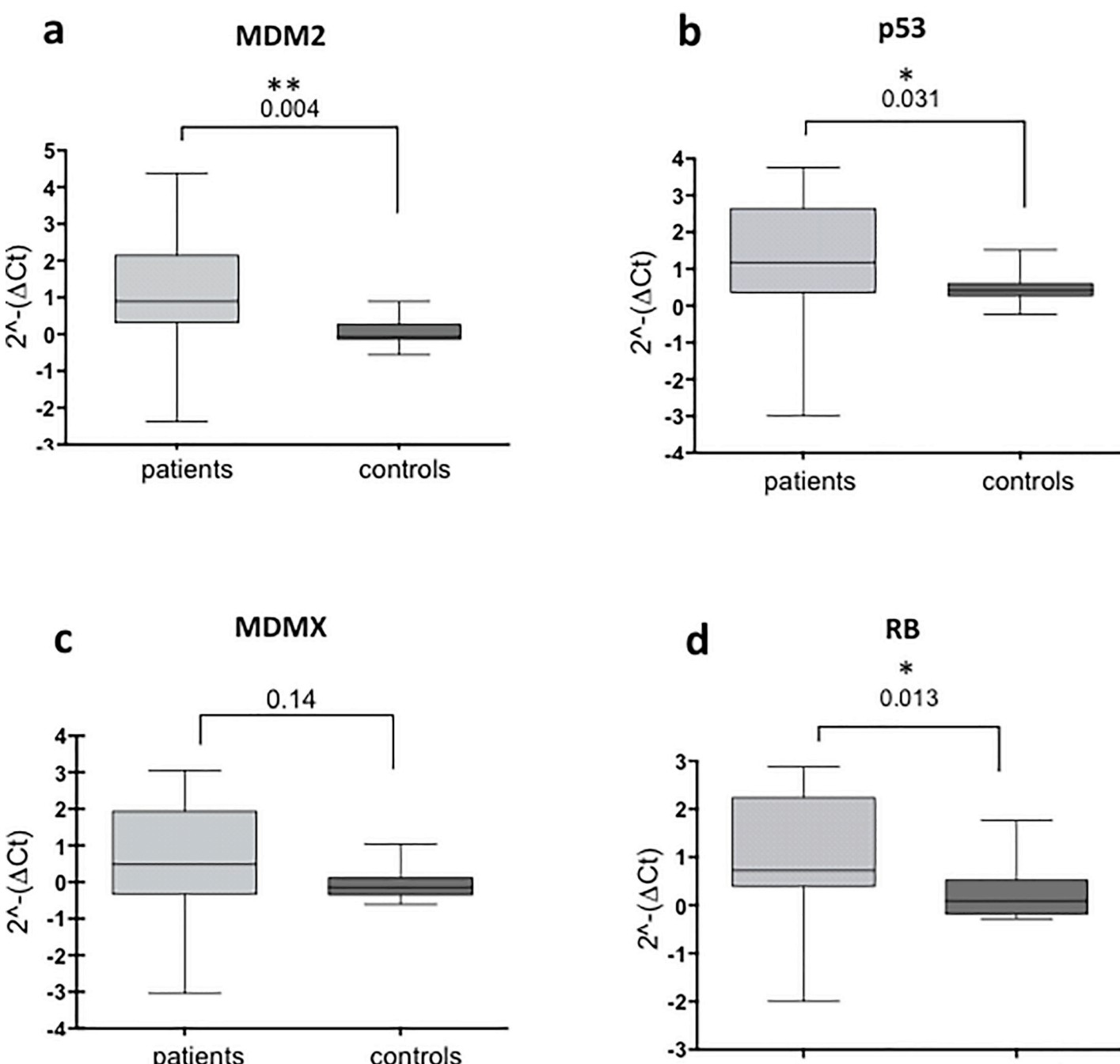

**Fig 1. Analysis of the mRNA expression levels in blood samples of controls and patients. (A)** Comparing the *MDM2* mRNA levels between 17 controls and 17 patients showed a significant difference between the groups. **(B)** Comparing the *p53* mRNA levels as in (A) showed a significant difference between the groups. **(C)** Comparing the *MDMX* mRNA levels as in (A), no statistically significant difference between the two groups. **(D)** Comparing the *RB* mRNA levels as in (A) showed significant difference between the control and patient groups. (p = *<0.05; **<0.01; ***<0.001).

## The p53, MDM2, MDMX and RB mRNA expression levels in peripheral blood of patients with retinoblastoma

It has been reported that the sporadic cases of retinoblastoma vary between 55–65%, with only 35–45% being hereditary. All the bilateral cases are thought to be hereditary, and between 10

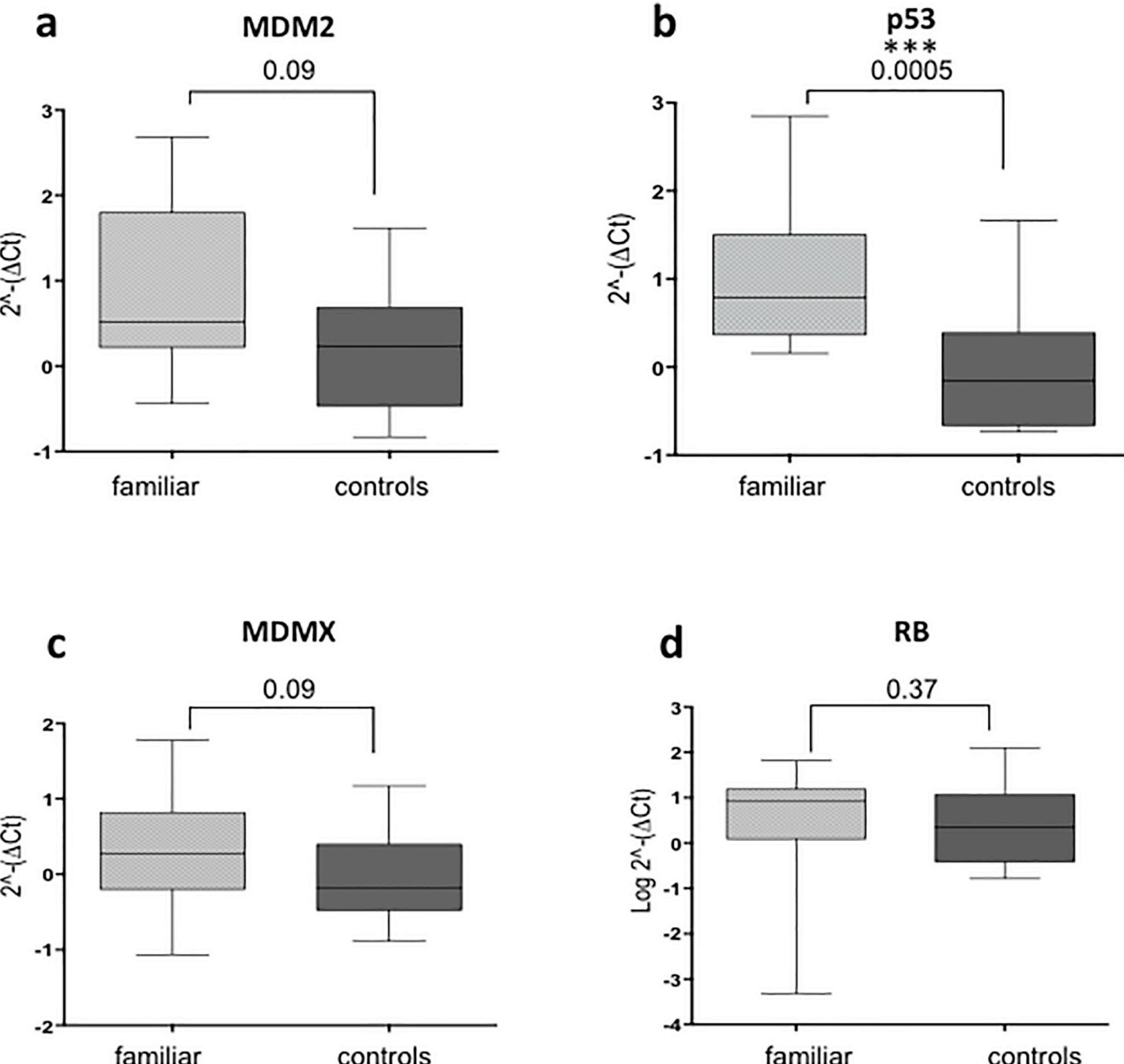

**Fig 2. Analysis of the mRNA expression levels in blood samples of controls and patients´ healthy family members.** (A) Comparing the *MDM2* mRNA levels between 20 heathy family members and 10 controls showed no significant difference between the groups. (B) Comparing the *p53* mRNA levels as in (A) showed a significant difference between the two groups. (C) Comparing the *MDMX* mRNA levels as in (A) showed no significant difference between the groups. (D) Comparing the *RB* mRNA levels as in (A) showed no significant differences between the groups. (p = *<0.05; **<0.01; ***<0.001).

and 15% of the unilateral forms of this cancer are also hereditary because one or more family members are affected [22,23]. We considered that the inherited form of the disease is underestimated, since RB mutations in somatic cells are rarely evaluated; furthermore, other genes could be altered. Based on these postulates, we wished to observe the levels of mRNA expression of the p53 pathway and RB in peripheral blood of patients with retinoblastoma. We

**Table 3. Correlations between patients´ characteristics and *p53*, *MDMX*, *MDM2* and *RB* gene expression.**

| Patients features | (%) | p53 | RB | MDMX | MDM2 |
|---|---|---|---|---|---|
| **Age** | | **0.04***  | 0.12 | **0.04***  | 0.09 |
| <1 | 41 | | | | |
| >1 | 58 | | | | |
| **Laterality** | | **0.01***  | **0.006**** | **0.01***  | **0.02***  |
| Unilateral | 63 | | | | |
| Bilateral | 37 | | | | |
| **Leucocoria** | 59 | 0.88 | 0.24 | 0.87 | 0.76 |
| **Sex** | | 0.18 | 0.92 | 0.18 | 0.19 |
| M | 59 | | | | |
| F | 41 | | | | |
| **Cancer family story** | 59 | 0.06 | 0.33 | 0.06 | 0.21 |

(*p < 0.05

**p < 0.01, ***p<0.001).

observed a significant difference in the levels of expression of *p53* and *MDM2* mRNAs in peripheral blood in patients compared with healthy controls. However, the levels of MDMX were unchanged. As we expected, the levels of *RB* mRNA also demonstrated a significant difference between controls and patients (Fig 1).

These results could be explained by an imbalance due to the cancer *per se*. A second option could be imbalance due to a hereditary characteristic. To investigate this, we examined the levels of expression of these genes in healthy family members of these patients. Strikingly, the results showed that only *p53* mRNA levels were significantly different in the healthy family members compared to the control group, while *MDMX* and *MDM2* mRNA levels show a considerable difference. The *RB* levels did not demonstrate any difference in expression level (Fig 2). These data suggest that indeed an hereditary p53 pathway factor could influence the development of retina cancer. Next, we decided to study the correlation of the levels of expression of these specific genes with some patients' characteristics.

## Association between *MDM2* and *p53* gene expression

We also investigated potential correlation between these specific genes and patient characteristics. We found an association between the patient´s age at diagnosis and expression of *p53* and *MDMX* (0.04) and to a lower extent with MDM2 (0.09); this is interesting because it has been postulated that the age of the diagnosis is associated with the laterality and inheritance of the disease. In agreement with the above data, the levels of expression of *p53*, *MDMX*, and *MDM2* mRNA are correlated with laterality (0.01, 0.01, and 0.02 respectively). The association was more significant with the *RB* expression level (0.006) (Table 3). We also investigated whether laterality and the age at diagnosis correlate with hereditary retinoblastoma via the correlation between the family´s history of cancer and the expression of these genes in family members. Surprisingly, we did not observe a strong correlation between these characteristics, just a weak association was observed with the expression of *p53* and *MDMX* mRNA levels and the cancer family story (0.06). It is important to note that some patients or their parents report being unsure about their family history of cancer. It should be noted that other characteristics including patient sex and some symptoms like leukocoria do not present any correlation with the expression of these genes (Table 3).

The p53 pathway genes in patients with retinoblastoma are, of course, related to each other. Spearman correlation shows that the p53 expression has a high correlation with MDM2 (0.89)

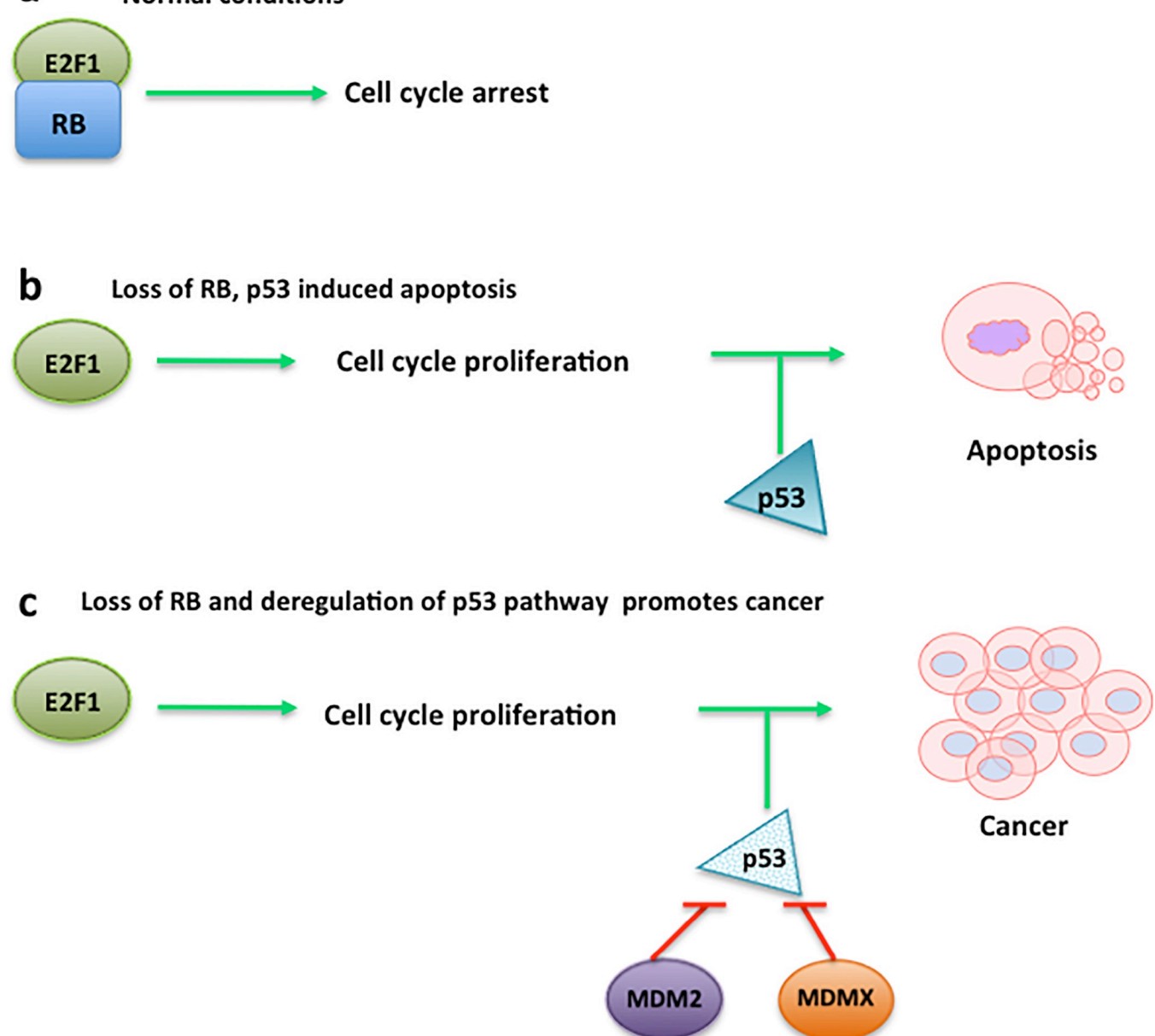

**Fig 3. Schematic representation of RB and p53 pathway in the development of retinoblastoma. (A)** In a healthy scenario of cellular conditions RB binds the transduction factor E2F1 to arrest the cell cycle. **(B)** Loss of RB could induce cell proliferation, however, the presence of p53 tumour suppressor will induce apoptosis of damaged cells, avoiding tumour development. **(C)** The loss of RB and the dysregulation of p53 pathway due to MDMX or MDM2 will promote cell proliferation and tumour development.

and a perfect correlation with MDMX (1). Nevertheless, the expression of *RB* is also positively correlated with p53 and MDMX (0.86) and MDM2 (0.84).

## Discussion

Cancer is a multifactorial disease; it involves sequential genetic lesions in the RB and p53 pathways [24]. Retinoblastoma arises from a cell that has lost RB, and while no injury of the p53 gene has been observed, dysregulation of MDMX or MDM2 has been reported [3, 10, 11].

Knudson (1971) observed that around the 40% of all the retinoblastoma cases are hereditary, meaning that one allele of an aberrant *RB* has been inherited from one of the parents and the other allele is inactivated by a somatic mutation. The additional 60% of cases are somatic or non-hereditary cases [22], meaning that during the development of the cones two somatic mutations occur. Nonetheless, the loss of RB function is not enough to develop tumorigenesis since it has been shown that the MDM2/MDMX/p53 pathway is a critical factor for the retinoblastoma (Fig 3). The young age of the retinoblastoma presentation suggests that the hereditary cases of the disease have been underestimated. If the role of p53/MDM2/MDMX pathway during retinoblastoma development has a hereditary factor, it could be found in any cell of the body. Then, blood is an exciting source of samples for the analysis of biomarkers. Here we investigate the p53 pathway in peripheral blood of patients with retinoblastoma, where results indicate that the level of the expression of *RB*, *p53*, and *MDM2* mRNAs are significantly altered in the samples of patients, but not the levels of *MDMX* mRNA. The levels of expression of these genes in the healthy family members of the patients demonstrated only a slight difference in *MDMX* and *MDM2* mRNA levels when comparing family members and controls; in the case of *p53* mRNA, the differences are significant between patients family member and healthy, unrelated controls, suggesting a hereditary factor is associated with the p53 pathway during development of the tumour in these patients. It was notable that, RB levels didn't present any change in these two groups.

The Spearman analysis shows that p53 is strongly correlated with MDM2 and MDMX, which is not surprising since one of the first genes transcribed for p53 is MDM2. Additionally, MDMX and MDM2 are responsible for p53 protein degradation [17]. RB and p53 are also strongly correlated, and it has been shown that RB⁻/RB⁻ mice die during the embryonic stages of the development [25]. This is due to p53 expression inducing apoptosis [26]. However, there is no evidence of compensation or directed genetic interaction between *p53* and *RB* until now.

It has been shown that the age of diagnosis is significantly lower in bilateral cases compared to unilateral cases [27, 28] and both, laterality and the age at diagnosis, are the principal features of inherited retinoblastoma. Our statistical analysis shows that the p53 and MDMX expression is associated with the age at diagnosis, and laterality correlates strongly with the RB gene expression but also with the p53, MDMX, and MDM2 levels. These correlations support a hereditary factor not only for the RB gene but also with the p53 pathway.

## Supporting information

**S1 Fig. Comparison of expression levels of candidate housekeeping genes in blood samples of retinoblastoma patients and healthy controls. (A)** A representation of electrophoresis gel confirming the amplicon size and primer specificity used in the RT-qPCR for all six housekeeping genes. **(B)** The boxes represent upper and lower quartiles of the cycle threshold range with medians. The grey boxes correspond to retinoblastoma patients. The white boxes represent healthy controls. **(C)** The differences of means and the matching symmetrical confidence intervals are shown for the relative expression of each housekeeping gene in the blood samples of patients and controls. If the symmetric confidence interval is included in the area of deviation and contains a zero, the gene is considered equivalently expressed between patients and controls.
(TIF)

**S1 Table. Housekeeping genes selected for expression analysis.**
(DOC)

**S2 Table. Statistical analysis of housekeeping level.**
(DOC)

**S3 Table. Stability ranking of the housekeeping genes.**
(DOC)

## Acknowledgments

We thanks to all patients and families.

## Author Contributions

**Conceptualization:** Martha Rangel-Charqueño, Vanesa Olivares-Illana.

**Formal analysis:** Mayra Martínez-Sánchez, Martha Rangel-Charqueño, Vanesa Olivares-Illana.

**Funding acquisition:** Vanesa Olivares-Illana.

**Investigation:** Jesús Hernandez-Monge.

**Methodology:** Mayra Martínez-Sánchez, Mariana Moctezuma-Dávila, Jesús Hernandez-Monge.

**Project administration:** Vanesa Olivares-Illana.

**Resources:** Martha Rangel-Charqueño.

**Validation:** Mayra Martínez-Sánchez, Mariana Moctezuma-Dávila.

**Visualization:** Mayra Martínez-Sánchez.

**Writing – review & editing:** Vanesa Olivares-Illana.

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
