## [Decision Letter · Decision Letter 0]

16 Apr 2020

PONE-D-20-08362

Analysis of the p53 pathway in peripheral blood of retinoblastoma patients; potential biomarkers

PLOS ONE

Dear Dra Olivares-Ilana,

Thank you for submitting your manuscript to PLOS ONE. After careful consideration, we feel that it has merit but does not fully meet PLOS ONE’s publication criteria as it currently stands. Therefore, we invite you to submit a revised version of the manuscript that addresses the points raised during the review process.

We would appreciate receiving your revised manuscript by May 31 2020 11:59PM. To enhance the reproducibility of your results, we recommend that if applicable you deposit your laboratory protocols in protocols.io, where a protocol can be assigned its own identifier (DOI) such that it can be cited independently in the future. For instructions see: http://journals.plos.org/plosone/s/submission-guidelines#loc-laboratory-protocols

We look forward to receiving your revised manuscript.

Kind regards,

Sumitra Deb, PhD

Academic Editor

PLOS ONE

2. Please provide additional details regarding participant consent. We note that you state "Written/signed informed consent was obtained from the guardians on behalf of the minors involved in the study". Please also state in the Methods and online submission information whether informed consent was obtained from all subjects in this study and whether it was written or verbal. If the need for consent was waived by the ethics committee, please include this information. For additional information about PLOS ONE ethical requirements for human subjects research, please refer to http://journals.plos.org/plosone/s/submission-guidelines#loc-human-subjects-research.

3. Please add the sources and catalog numbers of your reagents as well as details regarding all equipment used in your study to the Methods section of your manuscript.

4. We suggest you thoroughly copyedit your manuscript for language usage, spelling, and grammar. If you do not know anyone who can help you do this, you may wish to consider employing a professional scientific editing service.  

"This work was supported by Conacyt CB-256637. This work, including the efforts of

Vanesa Olivares-Illana, was funded by L´oreal-UNESCO-AMC. JH-M Cátedras

CONACyT program. We thanks to all patients and families."

Reviewers' comments:

Reviewer's Responses to Questions

**Comments to the Author**

1. Is the manuscript technically sound, and do the data support the conclusions?

Reviewer #1: Partly

Reviewer #2: Partly

2. Has the statistical analysis been performed appropriately and rigorously? 

Reviewer #1: Yes

Reviewer #2: Yes

3. Have the authors made all data underlying the findings in their manuscript fully available?

Reviewer #1: Yes

Reviewer #2: No

4. Is the manuscript presented in an intelligible fashion and written in standard English?

Reviewer #1: No

Reviewer #2: Yes

5. Review Comments to the Author

Reviewer #1: In this manuscript, the authors have attempted to monitor the status of the p53 pathway in peripheral blood samples obtained from retinoblastoma patients. They measure the mRNA levels of p53, RB, MDMX and MDM2 which could be further used as biomarkers for early diagnosis. Although it is an interesting premise and could be a good diagnostic approach, I am not able to understand the clear rationale behind utilizing the blood samples of RB patients to analyze the p53 pathway. The authors are requested to elaborate more on this. Moreover, other signaling pathways like the Wnt signaling is also highly functional in RB and could have higher significance as compared to p53. Have the authors looked into these pathways (mRNA analysis of beta-catenin etc)? Depending on the severity of RB, as mentioned, it could be unilateral or bilateral (simultaneously in both eyes) or in cases, it could start off as unilateral and at a later stage it could become bilateral. How could the presented approach address this issue? Also, peripheral blood would have circulating tumor DNA that could be coming from tumors that have metastasized from the retina to other parts of the body? In that case, would the diagnosis mentioned here delineate between the original vs metastasized tumor that could have a totally different p53 signature? The authors are recommended to discuss these in detail.

Optional: This may or may not be addressed. What about the more rarer cases of trilateral RB? Have the authors looked into the p53 signaling in these cases? If not, it would be good to include a few points about the same if possible.

Minor Changes:

1. Introduction...Line 1...Change "Paediatric" to "Pediatric".

2. The grammar needs to be formatted. There are lots of places where the sentences need corrections e.g. "it has been used to search of early biomarkers (Introduction)...should be changed to "it has been used to search or identify early biomarkers".

3. Proteasome 26 S needs to be written without any space.

4. I do not see the significance of writing about the housekeeping genes in detail. It could be just mentioned in a sentence or two.

5. The authors are requested to include these references:

a.https://www.ncbi.nlm.nih.gov/pubmed/16231976

b.https://www.ncbi.nlm.nih.gov/pubmed/15190215

Reviewer #2: In this manuscript authors have analyzed peripheral blood samples from patients suffering from retinoblastoma for mRNA levels of RB, p53, MDM2, MDMX and investigated the involvement of these genes in the development of cancer. There were some minor issues with the manuscript grammar and use of appropriate scientific language.

The authors need to provide complete profiling of the RB and p53 genes in the samples through sequencing and the status of mutations in them. The authors also show the comparison in the gene levels between patients, controls and healthy family in the same graph.

While the analysis shows some significant correlations between the p53, MDM2 and RB mRNA levels in patients and controls, some of the associations are not novel like that between p53 and MDM2. The sample size and the scope of analysis performed in this manuscript is not sufficient enough at this point to prove the use of these gene signatures as potential biomarkers.

6. PLOS authors have the option to publish the peer review history of their article (what does this mean?). If published, this will include your full peer review and any attached files.

Reviewer #1: No

Reviewer #2: No

---

## [Author Response · Author response to Decision Letter 0]

7 May 2020

Reviewer #1: In this manuscript, the authors have attempted to monitor the status of the p53 pathway in peripheral blood samples obtained from retinoblastoma patients. They measure the mRNA levels of p53, RB, MDMX and MDM2 which could be further used as biomarkers for early diagnosis. Although it is an interesting premise and could be a good diagnostic approach, I am not able to understand the clear rationale behind utilizing the blood samples of RB patients to analyze the p53 pathway. The authors are requested to elaborate more on this.

It has been shown that the p53 pathway is, indeed altered in retinoblastoma tumours, if this could be one of the causes of the diseases as has been proposed for several authors, then we thought that it could have a hereditary component, meaning that could be studied in any cell of the body. That was also the motivation to study the family members. We add more about it in page 4 and page 14 on the MS.

Moreover, other signaling pathways like the Wnt signaling is also highly functional in RB and could have higher significance as compared to p53. Have the authors looked into these pathways (mRNA analysis of beta-catenin etc)? 

The reviewer is right. Other pathways indeed could also be potential biomarkers for retinoblastoma. Has recently been published a review paper about it: 

Jie Sun , Hui-Yu Xi1,2, Qing Shao and Qing-Huai Liu. Biomarkers in retinoblastoma. Int J Ophthalmol, Vol. 13, No. 2, Feb.18, 2020. 

In the paper, the authors revised the potential biomarkers for retinoblastoma and both Wnt and p53 pathways, among many others, are mentioned. In this particular work, we focus on p53. 

Depending on the severity of RB, as mentioned, it could be unilateral or bilateral (simultaneously in both eyes) or in cases, it could start off as unilateral and at a later stage it could become bilateral. How could the presented approach address this issue?

This is an interesting question that we have been thinking about. We thought that a hereditary component of the p53 pathways is present. Then we could imagine that the bilateral cases would present higher significant differences in the levels of p53, MDMX or MDM2, more than the unilateral. However, this is difficult to demonstrate since, as the reviewer explains, a patient that at the moment of the study is unilateral, can later become bilateral. However, we found a correlation between the age of diagnosis and the levels of expression of p53 and MDMX m mRNA, and as has been postulated that the age of diagnosis in correlated with the laterality.

Also, peripheral blood would have circulating tumor DNA that could be coming from tumors that have metastasized from the retina to other parts of the body? In that case, would the diagnosis mentioned here delineate between the original vs metastasized tumor that could have a totally different p53 signature? The authors are recommended to discuss these in detail.

Of course, this is true, and it was our omission because we did not explain the inclusion/exclusion criteria in the MS. Now we add the following information on page 5 of the MS: 

Inclusion criteria: Children with the diagnosis of retinoblastoma, without a known diagnosis of other cancer. Exclusion criteria: Patients with a diagnosis of other types of cancer. Controls: Healthy children at pediatric age and healthy adults in the same range of age of the family member patients.

On the other hand non of the participating patient's in the study developed metastasis. Now we add a table with the characteristics of the patients in the MS.

Optional: This may or may not be addressed. What about the more rarer cases of trilateral RB? Have the authors looked into the p53 signaling in these cases? If not, it would be good to include a few points about the same if possible.

This is a good suggestion; unfortunately we were unable to obtain the sample of a patient with these characteristics.

Minor Changes:

1. Introduction...Line 1...Change "Paediatric" to "Pediatric".

Done

2. The grammar needs to be formatted. There are lots of places where the sentences need corrections e.g. "it has been used to search of early biomarkers (Introduction)...should be changed to "it has been used to search or identify early biomarkers".

Thank you, the correction is done and the MS has been carefully formatted

3. Proteasome 26 S needs to be written without any space.

Done

4. I do not see the significance of writing about the housekeeping genes in detail. It could be just mentioned in a sentence or two.

We reduced this part, and sent the Figure 1 (about the housekeeping genes) to Supplemental information.

5. The authors are requested to include these references:

a.https://www.ncbi.nlm.nih.gov/pubmed/16231976

b.https://www.ncbi.nlm.nih.gov/pubmed/15190215

Thank you for the suggestions, we added them!

Reviewer #2: In this manuscript authors have analyzed peripheral blood samples from patients suffering from retinoblastoma for mRNA levels of RB, p53, MDM2, MDMX and investigated the involvement of these genes in the development of cancer. There were some minor issues with the manuscript grammar and use of appropriate scientific language.

Done

The authors need to provide complete profiling of the RB and p53 genes in the samples through sequencing and the status of mutations in them. 

It is a good idea to get the mutation spectrum of RB and p53 in the patients. However, this is not the focus of the present work. In this manuscript we focus on the analysis of the levels of mRNA of the p53 pathways genes and RB. We will consider studying the mutation status of these genes for a future work. 

The authors also show the comparison in the gene levels between patients, controls and healthy family in the same graph.

While the analysis shows some significant correlations between the p53, MDM2 and RB mRNA levels in patients and controls, some of the associations are not novel like that between p53 and MDM2. 

The reviewer is right, the correlation between p53 and MDM2 it has been very well documented. Our results corroborates that in blood samples of retinoblastoma patients the correlation do not changes due to the disease. 

The sample size and the scope of analysis performed in this manuscript is not sufficient enough at this point to prove the use of these gene signatures as potential biomarkers.

We agreed; however, we hope that our work will stimulate others too and together, we can finally found a genes signature for this childhood cancer.

---

## [Decision Letter · Decision Letter 1]

26 May 2020

Analysis of the p53 pathway in peripheral blood of retinoblastoma patients; potential biomarkers

PONE-D-20-08362R1

Dear Dr. Olivares-Ilana,

We are pleased to inform you that your manuscript has been judged scientifically suitable for publication and will be formally accepted for publication once it complies with all outstanding technical requirements.

With kind regards,

Sumitra Deb, PhD

Academic Editor

PLOS ONE

Additional Editor Comments (optional):

Reviewers' comments:

Reviewer's Responses to Questions

**Comments to the Author**

1. If the authors have adequately addressed your comments raised in a previous round of review and you feel that this manuscript is now acceptable for publication, you may indicate that here to bypass the “Comments to the Author” section, enter your conflict of interest statement in the “Confidential to Editor” section, and submit your "Accept" recommendation.

Reviewer #1: All comments have been addressed

Reviewer #2: (No Response)

2. Is the manuscript technically sound, and do the data support the conclusions?

Reviewer #1: Yes

Reviewer #2: (No Response)

3. Has the statistical analysis been performed appropriately and rigorously? 

Reviewer #1: Yes

Reviewer #2: (No Response)

4. Have the authors made all data underlying the findings in their manuscript fully available?

Reviewer #1: Yes

Reviewer #2: (No Response)

5. Is the manuscript presented in an intelligible fashion and written in standard English?

Reviewer #1: Yes

Reviewer #2: (No Response)

6. Review Comments to the Author

Reviewer #1: The authors have addressed all the questions in the previous review. However, the figures seem to be pixelated. The authors are requested to check that.

Reviewer #2: (No Response)

7. PLOS authors have the option to publish the peer review history of their article (what does this mean?). If published, this will include your full peer review and any attached files.

Reviewer #1: No

Reviewer #2: No

---

## [Editor Report · Acceptance letter]

28 May 2020

PONE-D-20-08362R1 

Analysis of the p53 pathway in peripheral blood of retinoblastoma patients; potential biomarkers 

Dear Dr. Olivares-Illana:

I am pleased to inform you that your manuscript has been deemed suitable for publication in PLOS ONE. Congratulations! Your manuscript is now with our production department. 

With kind regards,

on behalf of

Dr. Sumitra Deb 

Academic Editor

PLOS ONE